# Agroecosystem resilience. A conceptual and methodological framework for evaluation

**Cindy Córdoba[1][ø], Catalina Triviño[2][ø], Javier Toro Calderón[iD][2][ø] ***

**1** Centro Interdisciplinario de Estudios sobre Desarrollo (CIDER), Universidad de los Andes, Bogotá, Colombia, **2** Instituto de Estudios Ambientales, Universidad Nacional de Colombia, Bogotá, Colombia

ø These authors contributed equally to this work.
* jjtoroca@unal.edu.co

**Data Availability Statement:** All relevant data are within the manuscript and its Supporting Information files.

**Funding:** This work received funding through the payment of salary as Associate Professor of the

## Abstract

This article proposes a conceptual and methodological framework for analyzing agroecosystem resilience, in which aspects such as agrarian structure and peasant community agency are included as determining factors. The methodology is applied to a comparison of two peasant communities in Latin America (Brazil and Colombia), emphasizing the capacity to transform unsustainable power structures in place of adapting to them. We find that when agrarian structure is more equitable and peasant agency is strongly developed through political formation, organization and women's participation, then there is a greater construction of resilience that improves peasant livelihoods and dignity. This application demonstrates that when agency is strongly developed, as in the case of Brazil, it is possible to transform structural conditions that restrict resilience. The inclusion and consideration of biophysical variables, management practices, agrarian structure and agency, through a participatory approach, allows for the identification of factors that inhibit or potentiate the resilience of agroecosystems.

## Introduction

The concept of resilience has evolved from an ecological perspective to that of complex systems analysis. Initially, it was conceived as the capacity to confront, absorb and adapt to disturbances, without changing, in order to return to a state of normality [1,2]. Resilience was calculated or evaluated depending on the amount of time it would take to return to this condition [3]. Analysis and discussions in the context of socio-ecological systems challenged the idea of normality, adopting an understanding of multiple equilibriums and accepting the inevitability of change [2,4]. In this sense, many proposed that resilience is systems adaptation based on learning, planning and reorganization for the purpose of preserving function, structure and identity [5–7].

Still, socio-ecological systems such as agroecosystems, conceptualized from a perspective of "fully integrated system[s] of people and nature" [8], do not exhibit unique identities, functions or structures [9]. Agroecosystems are systems composed of physical, biological, socioeconomic and cultural subsystems that coalesce and interact within the framework of human-led

Universidad Nacional de Colombia to Javier Toro Calderón, and salary as a post-doctoral researcher by the Interdisciplinary Center of Development Studies (CIDER) of the Universidad de los Andes for Cindy Cordoba and Der Deutsche Akademische Austauschdienst (DAAD) and The Center for Development Research (ZEF) is an institute of the University of Bonn, Germany for support of publication costs. The funders had no role in study design, data collection and analysis, decision to publish, or preparation of the manuscript.

**Competing interests:** The authors have declared that no competing interests exist.

agricultural processes [10,11]. In this sense, human intervention, expressed in different interests, values and criteria, impede the determination of a unique structure and system function [9,12].

Any system involving human interaction holds power relations that can form or influence resilience [13], since these determine which groups have access to and control of resources, assume the burden of risk, and have the possibility of participation and political decision-making [14,15]. Additionally, the fluctuating nature of these systems clashes with the concept of identity, which can be understood as seeking a static and invariable condition [16].

The complementary concept of resilience offered in this article is not necessarily neutral or inherently positive, due to the lack of consensus across society on the objectives and strategies for responding to or interacting with change or disturbances [17]. The resilience of agroecosystems is often power-dependent. While resilience can increase through the operation of privileged groups with greater access to resources and political participation, it can also decrease under groups with less economic power [18]. Therefore, it is necessary to question, resilience for whom and for what purpose? [16,19]. In this study, resilience is analyzed from the perspective of peasant and rural communities in Latin America. From the point of view of the elite, resilience is understood to be adaptation to conditions of inequality and injustice, which agrees well with neoliberal [20] and Keynesian discourse, in other words, maintaining the *status quo*. On the other hand, those with less power understand resilience to be transformation conducive of conditions of justice, which can lead to the destruction of the predominant social system [1,21–24].

To draw on the ideas of several authors [4,25–30], resilience is proposed as an emergent property of complex systems (family, region, nation) and as a result of the dialectical multi-level interaction of its elements. This property allows the system to buffer, to adapt to and most importantly, to innovate and **to be transformed**, not only in response to specific points of tensions, but also to continuous and inevitable biophysical and social changes in the environment. Resilience is not a neutral concept. It is lodged within power relations and should be analyzed considering point of view, interests, access to resources, and the researchers' place in society. Therefore, it does not only or mainly represent a system's return to a "normal" state. Rather, on the contrary, it necessarily implies discontinuities, structural changes and dynamic developments within the system.

The purpose of this article is to present a conceptual framework and complementary participatory methodology for analyzing and evaluating/measuring agroecological resilience, including factors relevant to agrarian structure and peasant community agency. We consider that these factors are determinants in building resilience, since greater degrees of peasant agency as well as equity in agrarian structure generate radical changes in social structures, allowing for the construction of resilience, and therefore generating conditions for the dignifying of the human condition. This approach and methodology are applied in the comparison of two rural peasant communities in Latin America (Brazil and Colombia), emphasizing the capacity to transform unsustainable power structures instead of adapting to them.

The first part of the article refers to the elements that are included in calculating resilience indicators, followed by an analysis of the reach and limitations of methodologies that have been applied in rural contexts. On this basis, a new methodology is proposed for analyzing resilience. This new methodology is then applied to two locations in Brazil and Colombia. The results are presented and discussed, followed by general conclusions.

## Agrarian structure

Whenever agroecological systems are analyzed, it becomes necessary to define the agrarian structure (AS), whose nucleus is the property of the land, based on which all other economic, social, cultural and political interactions are built. This concept combines a set of factors

including the size of agroecosystems, the use and control of resources, labor conditions, relationships among social actors and between social actors and the market, infrastructural aspects and other features [31].

Agrarian structure is understood as a territorial configuration that includes the way in which the entire territory is distributed and occupied [32,33]. From this formation, power relations are established as determined by development models that promote changes mainly in the size of the property, its concentration and / or fragmentation, as well as changes in land use [34]. [32], describes the following configurations of agrarian land structure:

- Unimodal:—Most of the land is distributed among medium-sized landowners (low concentration of property).

- Bimodal: Most of the land is distributed among a few owners, leaving a small proportion in the hands of many small owners (high concentration of property).

- Multimodal: Small and medium-sized property dominates the structure, coexisting in turn with large, highly use property. All these are articulated through alliances, business and cooperation systems (low concentration of property).

- In Latin America, land has been employed as an instrument of power and social domination [35,36]. High levels of land concentration (called "latifundia") or small subsistence-oriented (called "minifundia") farms constitute the principal motor for the backwardness and underdevelopment of the rural sector [37–40]. Since AS is transcendentally vital to productive power relations, peasant marginalization, territorial sovereignty, food production and access to dignified living conditions, it is surprising that it has not been included within analyses of resilience in the rural sector.

Land has historically been configured as a central means of production, whose appropriation and accumulation lay the groundwork for the construction of social power relations that determine the peasant population's access to resources, goods and services, is a main element of their dignity and identity, and defines a great extent of their autonomy, socioeconomic conditions and the development of their means of livelihood. Additionally, in small portions of land it is not possible to develop traditional cultivation methods, such as rotations, patches of vegetation, fallow areas, etc. All of these factors directly impact resilience and the capacity for transformation within rural communities [41–43].

Social dynamics determine how resources are distributed among social groups and therefore those who are most exposed to risk or are most vulnerable [44]. Identifying the structural causes of such vulnerability makes it possible to determine strategies to increase the resilience of rural communities, enabling their access to resources. This is what the livelihoods approach tries to do by including different types of factors. These incorporate **productive practices**, which includes criteria such as soil management and biodiversity that allow us to identify the ways in which agroecosystems deal with issues such as climatic variability. Another group of factors involves agroecosystem **conditions and context** that determine their degree of vulnerability. This group includes farmers' biophysical, social, and health criteria. Each criterion allows us to establish what resources communities have and where they stand in the face of continuously changing environmental conditions. The **agrarian structure** factor, although it has been mentioned by authors such as [44] as the most important asset for farmers, had not been quantified, nor have the criteria used for its specific analysis in agroecosystems been determined. Therefore, criteria such as productive relationships and land tenure reveal power relations that affect rural resilience and have been made invisible in agroecosystem studies. The last factor is the **capacity of agency** that will be developed in the next section.

The inclusion of this set of factors gives a more complete understanding of the agroecosystem within a framework of the asymmetries of power evidenced by each of the criteria. This approach allows for the simultaneous consideration of factors at multiple levels that shape resilience as a process, not only of adaptation but of transformation, towards conditions of greater dignity.

## Capacity of agency

The capacity of agency is understood as the empowerment of marginalized communities to engage in collective objective-oriented action aimed at transforming societal power relations [24,45,46]. Agency goes beyond resisting, buffering or adapting to the hardships of capitalism. It implies that peasants can build new paths in response to a system they consider unsustainable [2], employing their own creativity, political decision-making and organizational power, to unravel their own development processes.

The capacity for transformation that stems from community agency allows for access to resources such as land, favoring food sovereignty and increasing true participation in political decision-making processes, leading to greater resilience. On the contrary, the capacity for adaptation could give way for a community to return to a previous state of injustice and inequality, without questioning the asymmetries of power [47]. Resilience is developed when communities construct new paths, reorganizing ecological and social structures and redistributing risks, costs and benefits to produce social transformations.

The role of organized collective agency has not been sufficiently integrated within analyses of resilience [46]. More research is needed to include the ways in which human actions shape factors such as agrarian structure. Resilience analyses in agroecosystems have not considered power relations, assuming the existence of a society in consensus, in which it is common for certain groups to support the disasters provoked by capitalism [17,48]. Nevertheless, it is relevant to include the way in which conscious choices made collectively and individually can transform conditions of inequality towards essentially new systems, this being a fundamental factor in the level of resilience acquired [9,49,50].

However, some authors have tried to integrate factors such as power, inequality and agency into resilience thinking. [44,51–53] argue that resilience has been used to reproduce neoliberal discourses, reducing state participation, while promoting community self-sustainability and individual responsibility. In this way, hegemonic thoughts and values are normalized and multiplied.

Contrary to the idea of resilience as a return to a state of normalcy or adaptation to unfair social conditions, another approach raises resilience as a transformation of social and environmental problems, based on forms of community activism and organization [46,54]. Articulating resilience with the fundamental problems of capitalism is still a field to explore at a theoretical and empirical level [51].

[55] On the other hand, points out that although conflicts, inequality and power relations are part of human societies, there is a generalized assumption that there is consensus among social groups. This ignores unequal exchanges of technology, information and resources in the predominant development model. Therefore, the concept of socio-ecological resilience must address the historical problems of conflicts of interest, asymmetries of power and inequality in distribution [24,25]. However, it would not be logical for capitalism to admit or promote its antithesis, that is, a definition that questions it in its structural aspects.

Sectors with power in society seek to create a generalized approach to sustainability, which marginalizes and excludes populations with less political and economic power. In this way, the latter forge a path through citizen action and social movements to exercise governance based on principles of learning, participation and trust [56].

In the case of agroecosystems, it is necessary to consider that beyond the biophysical space they occupy, they are also influenced by symbolic, economic, political and technological relationships that interact on multiple scales and hierarchies [57]. Therefore, agroecosystems also reflect disputes over natural and economic resources and are affected by social struggles.

In this sense the proposed methodology for evaluating resilience includes the decisions peasants make about the use of resources for agricultural production (both infrastructure and subsistence), as well as the level of organization, training and political decision-making power [18,58]. It is relevant to incorporate a differentiated analysis, not only of the economic situation of women (pay for market-oriented work, subsistence and caregiving), but also of aspects related to their empowerment, such as the levels of organization and participation in political decision-making processes [35]. The participation of women is essential because they are considered to be political subjects who organize and participate in decision-making regarding economic, productive, technical and political aspects, thus transforming power relations [59–63]. This set of factors allows farmers not only to endure any type of stress (economic, political, ecological, etc.), but to act on these factors to transform them in their favor.

## Methodologies for evaluating resilience

There are many methodological problems and few evaluation frameworks for resilience in rural contexts [49,57,64]. Some methodologies are centered around ecological and productive variables, employing indicators such as landscape complexity, vegetation diversity, slope and soil conservation, energy efficiency, subsistence, water and soil conservation practices, input and technology dependence and others [10,65–69]. These approaches address social factors only in a limited and tangential way through their general definition of resilience as the capacity of communities to adapt to extreme stressors within the productive sector.

Authors such as [4,46,49], recognize that the social aspects of resilience are weakly developed, especially with regards to empowerment. [54,70] include notions of collective community agency as important to resilience, but they do not propose measurement instruments. [71] presents eight [8] dimensions of community resilience with metrics that have not been applied in practice and that are centered on the capacity to adapt to change. [72] employ official statistics to propose an index of rural diversity, considering natural economic and social capital, under the premise that diversity increases resilience. Other authors include, in addition to ecological variables, factors such as food security, income, access to services and support networks [73]. These are, however, included without numerical qualifiers or variable weights. Although [74] quantify variables such as land size, financial sources, credit and network participation, these are limited to describing the way in which these influence the adoption of agricultural technologies. [57] present 13 indicators of agroecosystem resilience which include social organization, learning, local knowledge and autonomy. Nevertheless, none of these variables consider social inequalities or access to land, determining factors for peasant livelihoods. [75,76] Introduce aspects such as social inequality and land property, recognizing that the socio-cultural context limits resilience. They center their attention on the capacity of farmers to respond with productive agroecological practices and define empowerment as decision-making for adaptive farm management in response to disturbances.

None of these studies includes the role of peasant agency in the transformation of structural factors that subvert power relations, bypassing the role of political organization and the building of new pathways, not only in the productive or ecological sense, but also in the social and political spheres. Productive relationships, working conditions and the use and control of resources are not evidenced, neither is it specified what social group's perspective is being analyzed in terms of resilience. All of this leaves unanswered the questions of resilience for what

end? and for whom? raised by [16]. Authors that consider the transformation of the status quo instead of its preservation [2,70], do not develop methodological proposals for the quantification of principal variables.

Resilience is the result of complex interactions among ecosystems, economic, social and cultural systems and cannot be analyzed through a fragmented consideration of each component in isolation from the whole [72]. With this challenge in mind, a methodology is proposed for measuring resilience in rural peasant communities, through the quantification and weighing of differing attributes. In addition to aspects related to AS and peasant agencies, related factors are incorporated to the conditions and context in which productive activities are developed, including biophysical, social and health variables, as well as practices used in agricultural production. In addition, market interactions were considered, which represent the effect of variables out of the peasants' control that exercise a strong impact on income level and livelihood development.

Therefore, it is necessary to present a complementary conceptual and methodological framework that allows the identification of factors that support or inhibit resilience in Latin American peasant communities. The complex analysis of diverse factors that constitute resilience, with an emphasis on AS and the capacity of collective agency, allows for an understanding of substantial aspects in need of transformation. This allows peasants to generate their own development dynamics based on their own interests and needs, favors processes of empowerment for implementing radical changes in the generation of public policy, access to resources and capital, and potential for autonomy [77–79]. In this sense resilience refers to social change and challenges the *status quo* to give place to alternative scenarios [1,22,23].

Resilience cannot necessarily be measured after specific events that cause shock or stress, since when dealing with complex systems that are subject to constant change and fluctuation, it is possible to identify their degree of resilience at any time. In addition, not every change or alteration is negative, but can lead to alternative scenarios and new directions desired by the farmers. Resilience can be measured to identify crucial points needed to achieve structural transformations through public policies, management plans and demands for social struggle.

## Material and methods

The community of peasants interviewed in the research was involved in the planning and development of the methodology, as well as in the analysis and discussion of the results. The ethical procedure included informed consent on the use of their testimonies in interviews and surveys. This involved specifying that the use of the information has an exclusively academic purpose and that they could stop participating at any time. The peasants contributed and benefited from the research by participating and discussing the study topics. The methods applied in this research and the peasants and professionals interviewed were approved and a endorsed by the National University of Colombia, Faculty of Agricultural Sciences, in the framework of the Ph.D. Thesis "Resilience and Climate Variability in Coffee Agroecosystems in Anolaima (Cundinamarca, Colombia)" (Record/Acta No. 013, 28[th] May 2012, Council of the Faculty of Agricultural Sciences, National University of Colombia).

### Proposed methodology for evaluating resilience

The procedure for evaluating resilience consists of three phases: (i) selection and weighting of factors, criteria and variables, (ii) scoring of variables, (iii) assigning quantified values to resilience.

### Selection and weighting of variables

A scoring matrix was built with a hierarchical structure composed of four [4] factors, eight [8] criteria and seventeen [17] variables (Fig 1). Weighting coefficients were assigned to each

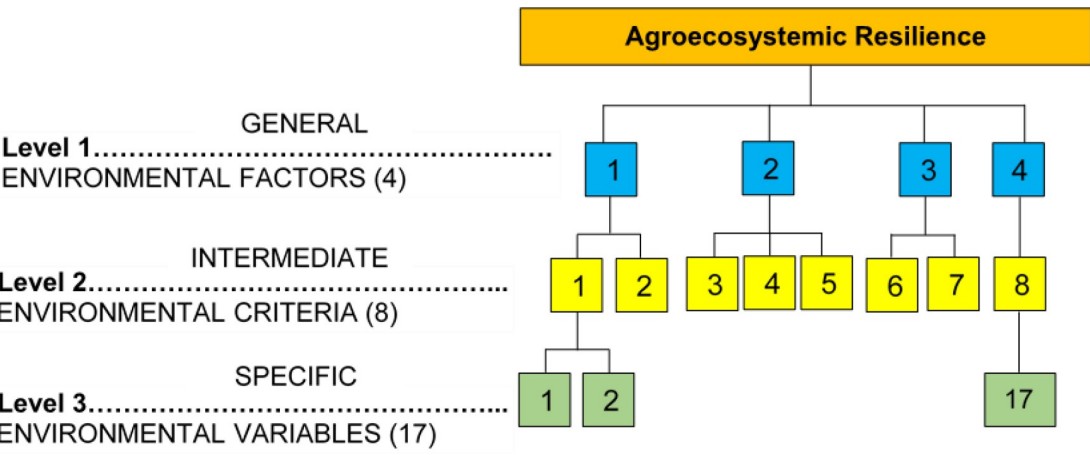

**Fig 1. Hierarchical structure for the evaluation of agroecosystem resilience.**

variable, factor and criteria, through consultation with principal actors in each community as well as expert opinion from several disciplines (anthropology, agroecology, health sciences, environmental sciences and administration). The final values were determined in a participative manner using the Delphi method (Table 1), which establishes a structured communication between experts and community members who are knowledgeable of study sites, to validate each category used in the analysis [80,81].

This methodological approach is mixed methods and participatory, since communities and local actors are consulted with to determine both qualitative and quantitative values of factors, criteria and variables. This characteristic allows the methodology to be applied or reproduced

**Table 1. Weighting matrix of factors, criteria and variables for the assessment of resilience.**

| Factor | Criteria | Variables |
|---|---|---|
| **Capacity for agency [50%]**\* | **Political-organizational [70%]** | Pertinence and/or link to organizations, cooperatives, and educational institutions [**11.67%**] |
| | | Level of training and political decision-making power [**11.67%**] |
| | | Level of training and political decision-making power (women) [**11.67%**] |
| | **Use of resources [30%]** | Subsistance (animal and vegetable) \* [**7.5%**] |
| | | Infrastructure [**7.5%**] |
| **Agrarian structure [37%]** | **Land tenure [50%]** | Property size\* and/or area [**9.25%**] |
| | | Land ownership [**9.25%**] |
| | **Production relationships [50%]** | Labor conditions [**6,17%**] |
| | | Market relationships [**6.17%**] |
| | | Level of income\* [**6.17%**] |
| **Conditions and context [10%]** | **Biophysical factors [25%]** | Soil quality [**1.25%**] |
| | | Distance to forests and water sources [**1.25%**] |
| | **Social factors [45%]** | Access paths [**2.25%**] |
| | | Access to public services and telecommunications [**2.25%**] |
| | **Health factors [30%]** | Drinking water [**1.5%**] |
| | | Frequency of protein consumption [**1.5%**] |
| **Productive practices [3%]** | **Soil management and biodiversity [100%]]** | Soil management and biodiversity [**3%**] |

[\*] Values within brackets are proposed weights.

in other territorial contexts, because the variables are evaluated according to the characteristics of the study site and are qualified according to generic scores applicable in various rural contexts (Tables 1 and 2). The weights are adapted to the criteria of local actors that participate, whether indigenous, black or afro communities or others.

## Scoring of variables

Data for differing variables were reported in different measurement units. For example, the area of land is expressed in hectares and the level of income in currency, while other characteristics are qualitative (land ownership or pertinence to organizations). Therefore, all measurement units were transformed to a standard 0 to 5 scale, where 0 represents the lowest level of resilience and 5 the greatest. This methodological strategy has been utilized and validated in several similar studies [82–90]. The values were negotiated in a participatory manner, employing questionnaires, semi-structured interviews, expert opinion and literature review. Table 2 presents the consolidated matrix with scoring criteria.

## Quantitative assessment of resilience

The value of agroecosystem resilience is the result of the sum of the 17 weighted variables. Where: AgRe: Agroecosystemic Resilience; Vi: Variables; Wi: Weight.

$$AgR_e = \sum_{i=1}^{i=17} v_{i*Wi} \qquad (1)$$

## Application

The proposed methodological model was applied to two localities in Colombia and Brazil: the municipality of Marulanda within the state of Caldas in Colombia (Lat 5˚ 17' 3" North, Long 74˚ 15' 48" West), and the municipality of Varzelandia within the North of Minas Gerais in Brazil (Lat 15˚ 42' 5" South; Long 44˚ 1' 39" West) (Fig 2). These sites were chosen because they share certain aspects such as the bimodal structure of land ownership, where "latifundia" and fragmented smallholder farms are predominant, with self-sustainable agricultural family units (Family Agricultural Units or UAF) and inspection units (Fiscal Modules) under the recommended area (18,83 ha) by the Colombian Institute for Rural Development (Colombian Institute for Rural Development or INCODER) in Colombia and the recommended area (50 ha) by the National Institute for Colonization and Agrarian Reform (INCRA) in Brazil. This land concentration generates inequality in power relations, that should be considered when measuring resilience in rural communities. On the other hand, the marked differences between these two communities to transform their socioecological systems allows a comparison of their level of agency and how this influences the final evaluations of resilience in each case.

The municipality of Marulanda has an area of 37,857 hectares and is divided into 14 neighborhoods. Soils are predominantly susceptible to erosive processes and high slopes are prevalent. There is a wide network of rivers and streams, an average temperature of 19˚C and an altitudinal range between 1550 and 3750 meters above sea level (ms.asl).

In this municipality, which has an approximate population of 2,081 people [91] and whose Family Agricultural Unit (UAF) is defined at 18.83 hectares, the base of the economy in the agroecosystems located at an average altitude of 1800 ms.asl is the cultivation of coffee, sugarcane cane and avocado. On the other hand, agroecosystems located at an average height of 2800 ms.asl rely mainly on the livestock sector through dairy farming and cattle raising. There is also a small number agroecosystems dedicated to sheep raising for meat and wool [92].

**Table 2. Resilience scoring matrix.**

| Factor: Capacity for Agency | | | | |
|---|---|---|---|---|
| **Criteria** | **Variable** | **Question** | **Answer** | **Score** |
| Organizational-political | Pertinence and/or link to organizations, cooperatives, and educational institutions | Do you pertain to or are you linked to an organization that. . ..? | Favors the capacity of economic and political transformation of the community, favors the capacity for transformation of the agroecosystem. | 5 |
| | | | Favors the capacity of transformation at the agroecosystem level. | 3 |
| | | | Generates little or no betterment of resilience conditions. | 1 |
| | Level of training and political decision-making power | What is the level of participation in community decision-making processes (regarding technical, productive, economic or political decisions)? | High | 5 |
| | | | Medium | 3 |
| | | | Low | 1 |
| | | What is the level of participation in political training meetings aimed at learning about and demanding rights? | High | 5 |
| | | | Medium | 3 |
| | | | Low | 1 |
| | Level of training and political decision-making power (women) | What is the level of participation and political organization of the women in the neighborhood or municipality? | High | 5 |
| | | | Medium | 3 |
| | | | Low | 1 |
| | | | Does not participate/ there is no organization | 0 |
| Use of resources | Subsistence (animal and vegetable) * | Number of animal species produced on the farm and used for subsistence | Two standard deviations above the average | 5 |
| | | | Two standard deviations below the mean | 0 |
| | Infrastructure | How do you rate the installations, tools for production and irrigation (if necessary) used for your main economic activity? (taking the mean of the three variables) | Very good | 5 |
| | | | Good | 5 |
| | | | Average | 3 |
| | | | Poor | 2 |
| | | | Very por | 1 |
| | | | Does not possess infrastructure | 0 |
| **Factor: Agrarian Structure** | | | | |
| Land Tenure | Size of land* | Area of the farm in hectares | If the size of the land> = UAF, then the score is 5, otherwise the score is calculated as (size/UAF) *5. | 0–5 |
| | Land ownership | Type of property | Landless | 0 |
| | | | Sharecropper | 1 |
| | | | Renter | 2,5 |
| | | | Owner (with land title) | 5 |
| | | | Owner with land title from a peasant organization | 5 |
| | | | Collective property | 5 |
| Production relationships | Labor conditions | Labor rights: Is there an established work schedule, rest period, vacation time and endowments? (averaging the 4 factors) | Yes | 5 |
| | | | No | 0 |
| | | Do you participate in any collective productive activity in your community? | Yes | 5 |
| | | | No | 0 |
| | | Paid family labor (principal product) | Always | 5 |
| | | | Occasional | 3 |
| | | | Never | 0 |
| | | Compensation for women for jobs such as: sustenance, domestic responsibility, production for the market (averaging the 3 factors) | 3 jobs | 5 |
| | | | 2 jobs | 3 |
| | | | 1 job | 1 |
| | | | Never | 0 |
| | Market relations | What is the level of decision-making power regarding product market prices? | Medium | 3 |
| | | | Low | 1 |
| | | | Nonexistent | 0 |
| | Level of income* | What is your average level of income? ** | Under minimum wage (MW) | 0 |
| | | | (Income*5) /2 MW | 3 |
| | | | Over or equal to 2 MW | 5 |

(*Continued*)

**Table 2.** (Continued)

| Factor: Capacity for Agency | | | | |
|---|---|---|---|---|
| **Criteria** | **Variable** | **Question** | **Answer** | **Score** |
| **Factor: conditions and context** | | | | |
| **Biophysical factors** | Soil quality | How do you rate soil fertility on your farm? | High | 5 |
| | | | Medium | 3 |
| | | | Low | 1 |
| | | | Not fertile | 0 |
| | | Gradient on the farm | None 0° | 5 |
| | | | Very low 0%-5% (0–8,5°) | 4 |
| | | | Low 15%-30% (8,5°-16,7°) | 3 |
| | | | Medium 30%-50% (16,7°-26-6°) | 2 |
| | | | High 50%-100% (26,6°-45°) | 1 |
| | | | Very high >100% (45°) | 0 |
| | Distance to forests and water sources | Distance of the agroecosystem to natural forest fragments (using area geometry and spatial analysis) | High: between 0 and 300 meters. | 5 |
| | | | Medium: between 300 and 500 meters. | 3 |
| | | | Low: between 500 and 1.000 meters. | 0–1 |
| | | Distance of the agroecosystem to bodies of water (using area geometry and spatial analysis) | High: between 0 and 50 meters. | 5 |
| | | | Medium: between 50 and 100 meters. | 3 |
| | | | Low: between 100 and 300 meters. | 0–1 |
| **Social factors** | Access paths | Principal access path from the farm to a point of sale for the main product | Paved road | 5 |
| | | | Combined paved road and unpaved road | 4 |
| | | | Unpaved road | 3 |
| | | | Trail | 2 |
| | | | Bridle path | 1 |
| | | | No access paths | 0 |
| | Access to public services and telecommunications | Public services (drinking water, light, household gas) | All 3 | 5 |
| | | | 2 of 3 | 3,3 |
| | | | 1 of 3 | 1,7 |
| | | | None | 0 |
| | | Communications (newspaper, telephone (cellphone signal), internet, radio, tv) | All 5 | 5 |
| | | | 4 of 5 | 4 |
| | | | 3 of 5 | 3 |
| | | | 2 of 5 | 2 |
| | | | 1 of 5 | 1 |
| | | | None | 0 |
| **Factor: conditions and context** | | | | |
| **Health factors** | Drinking water | Do you have access to clean drinking water? | No | 0 |
| | | | Si | 5 |
| | Frequency of protein consumption | Number of protein products consumed daily by every member of the family (eggs, legumes and meats) | (# times a week) /21) *5 | 0–5 |
| **Factor: productive practices** | | | | |
| **Soil management and biodiversity** | Soil management and biodiversity | Do you use polyculture or accompanying diversity for pest control, increased soil fertility or subsistence agriculture? | No | 0 |
| | | | Yes | 5 |
| | | How often do you use herbicides, pesticides and synthetic fertilizers? | High | 0 |
| | | | Medium | 1 |
| | | | Low | 3 |
| | | | None | 5 |
| | | How would you rate your level of traditional knowledge and/or training in agroecology? | High | 5 |
| | | | Medium | 3 |
| | | | Low | 2 |
| | | | None | 0 |

*Data were normalized, and atypical values were eliminated, then the mean and standard deviation were calculated.

**Minimum wage salary for Colombia is: 264,67USD, and for Brazil: 264,58 USD.

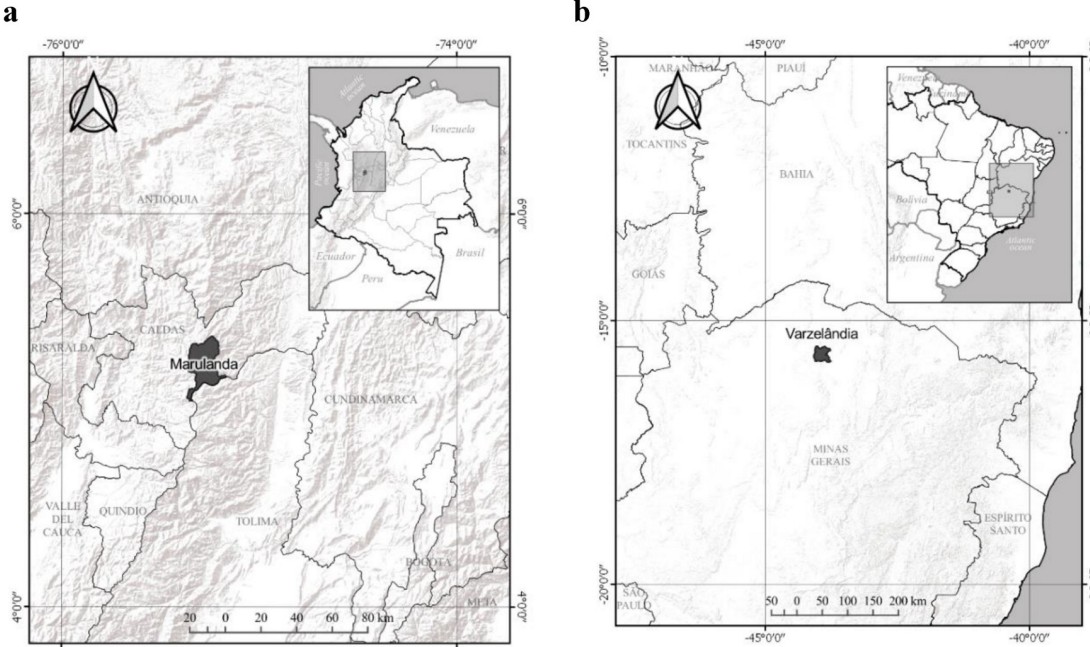

a. Municipality/Town of Marulanda, Departament/State of Caldas, Colombia
b. Municipality/Town of Varzelandia, Department/State of Minas Gerais, Brazil

**Fig 2. Localization of study areas.**

Varzelandia has an area of 81,499 ha, an altitude of 745 meters above sea level, an average temperature of 29˚C and an estimated population of 19320 inhabitants [93]. The area is predominantly dry forest which has been diminished due to the extensive livestock activities that occupy almost 77% of the municipality. These large agroecosystems dedicated to livestock belong to 8 landowners [94].

The agroecosystems that occupy the remaining percentage of the municipality are allocated to subsistence agriculture with corn, beans and sugarcane plantations as well as small-scale livestock production. The production is for self-supply. Many of the surplus products are sold internally and others in the urban area of the municipality. The main product that is transformed is cassava flour and in some productive units, panela is produced [94].

## Data collection

Qualitative and quantitative methods were combined for the analysis of biophysical and sociocultural conditions that come into play in the resilience of both communities. The following data collection instruments were used:

**Participatory workshops.** 5 group workshops were conducted in the municipality of Marulanda and 8 in the municipality of Varzelandia, including main actors in each municipality. In the workshops variables and resilience scoring criteria were defined in a participatory manner.

**Surveys.** surveys were conducted in each of the studied agroecosystems (N = 34), employing a questionnaire composed principally of close-ended multiple-choice questions and forecasting [95].

**Semi-structured interviews.** 23 semi-structured interviews were conducted in Marulanda and 31 in Varzelandia, with town officials, peasants, leaders of political and local organizations, which permitted a greater degree of flexibility and depth in obtaining information [96]. The interviews were conducted in different workspaces of planting and harvesting, local commerce and the home.

## Results, discussion, conclusions

It is necessary to clarify that resilience was measured at the present time, the way in which each community arrived at its current state of resilience state was analyzed and qualified, as well as the main factors that led them to this point and that determined the degree of resilience of each location.

In consensus, the communities of both municipalities and experts assigned a coefficient of 0.5/1.0 to the capacity for agency, since it represents an indispensable factor for the construction of resilience. Agency is directly related to the ability of the community to self-organize and strengthen autonomy and participation in decision-making spaces, generating transformations, adjustments and modifications at different scales in each social, economic, political, ecological, and livelihood context.

The factor that was given the second most important weight was agrarian structure (0.37/1.0), which consists of the size of the agroecosystem, the type of ownership and other factors derived from the first two, such as market relations, working conditions and income level. The remaining criteria, "conditions and context" and "productive practices", were given lesser relevance in the construction of resilience, since they can be modified by human agency. Therefore, they were assigned a weight of (0.1/1.0) and (0.03/1.0) respectively.

Fig 3 shows that the resilience of agroecosystems in the municipality of Marulanda, Colombia is lower than that of agroecosystems in the municipality of Varzelandia, Brazil. The municipality of Marulanda had low scores (< 2.2), while the municipality of Varzelandia had scores between 1.9 and 3.5.

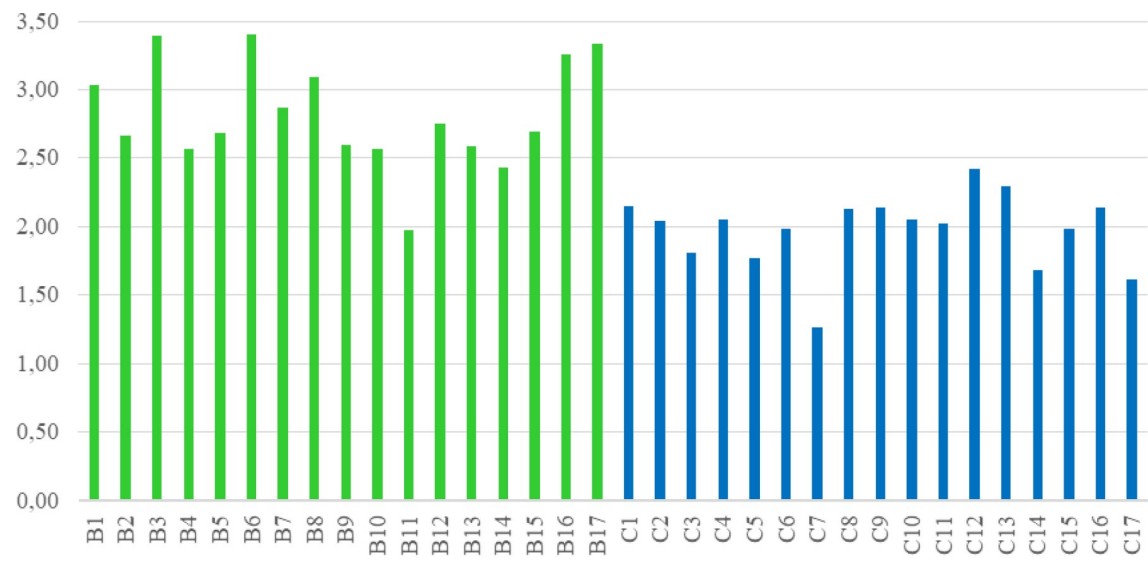

**B**: Brazil. **C**: Colombia

**Fig 3. Total values of resilience in Brazil and Colombia.**

The analysis of variance showed, with a confidence level of 95%, that there are statistically significant differences between the resilience of the municipalities of Varzelandia and Marulanda. Also, the resilience of agroecosystems in Varzelandia is significantly greater than the resilience of agroecosystems in Marulanda, and that this result is not due to chance [97]. Variables with statistically represenative differences include: the degree of membership in organizations, the degree of training and political decision-making power, political participation of women, infrastructure, land ownership and working conditions.

The inclusion of AS and agency criteria allows for a closer representation of reality and explains why some variables held higher ratings than others. In the case of Varzelandia, the peasants' capacity for agency modified certain aspects related to agrarian structure, for example, through the occupation and ownership of fertile lands (average score of 4.6) and flat lands (average score of 4.3), which were previously owned by powerful landowners (77% of the territory was held by 8 landowners). This factor also allowed for a transformation of productive relationships by developing a collective production area where women, youth and elderly are remunerated through hourly pay (average score of 2.2 versus 0.4 for the municipality of Marulanda). Prior to developing the collective area, the peasants in Varzelandia worked under local landowners and were often exploited. This implies that beyond adapting, they managed to transform structural conditions, enhancing their resilience.

In addition, the capacity of organization and community-level management created enough pressure for the Mayor and city council to provide materials and machinery for the construction of a deep well and bridge over the river Arapuim, thus improving the infrastructure score (3.6 versus 2.8 for the municipality of Marulanda). In addition, the community committed itself to facilitating labor for these two projects, carrying out the process collectively. The installation of the deep well guarantees irrigation for the collective production area, and the construction of the bridge improves connectivity, transport and quality of life.

On the contrary, in the municipality of Marulanda, the community adapted to social conditions without achieving transformations that would improve peasant livelihoods. Therefore, in general, the score for pertaining to or connecting with organizations or cooperatives (average 2.4), as well as the degree of training and family-level political decision-making power (average 1.0) and especially women's decision-making power (average 1.0), was low in all cases. In this municipality, the peasant smallholder has restricted access to resources, goods and services, and productive activities use unpaid family labor intensively, in order to increase their precarious income and improve living conditions. It is evident in this case that simple commodity production, developed individually, limits the accumulation of capital [98,99].

Socio-economic conditions influence community agency. In the municipality of Varzelandia, the deep history of land struggles and strong peasant organization has allowed farmers to solve problems related to land tenure and production relationships. However, in the municipality of Marulanda, the historical absence of land struggles has maintained a limited division of land parcels through informal agreements, perpetuating the dominant economic position of powerful landowners [98].

Factors at all scales affect the resilience of the agroecosystem. For example, peasants have no impact on the prevailing factors governing market relations, and therefore, market relations are scored as zero in both municipalities, regardless of the capacity for agency. The fixing of product prices is determined by various dynamics of the capitalist market and by local economic powers [100].

Dependence on the country's agricultural policies or international fluctuation of prices negatively affect resilience [2,101]. Therefore, it is necessary to include power relations derived from global scales, which prevent peasants from reaching full autonomy in decision making or real participation in processes of political definition [8,16,19,102].

Weighing criteria and landscape indicators give a closer sense of the reality of the case studies and allows a greater understanding of the factors that most strongly affect resilience. Without weighing variables, certain factors such as soil fertility, slope or access to public services would be considered on an equal level as criteria related to community agency or agrarian structure. The proposed methodology includes aspects that are normally invisible, revealing power relations and transformation processes that alter structures and predominant social dynamics within communities [21,103].

Fig 4 shows the results of calculating resilience without considering AS or agency, utilizing criteria associated with productive practices and biophysical conditions in comparison with the weighted average using all the proposed variables. The results of the municipality of Marulanda are higher in scenario Y than scenario X, with a variance between 3% and 74%. On the contrary, the municipality of Varzelandia showed lower results for scenario Y, lowering the mean values of resilience, with soil fertility and slope being the variables with the greatest weight.

When only biophysical factors and agricultural practices are considered, there is only a difference of 0.2 points between the two localities (an average score of 2.2 for Colombia vs. 2.4 for Brazil). On the contrary, when all variables are considered, the difference between average values is almost a whole point (an average score of 1.8 for Colombia vs. 2.8 for Brazil).

The values with the highest scores in the municipality of Marulanda were distance to forests and water sources (average 4.4 vs. 1.0 in Varzelandia) as well as the presence of rivers and water sources within ecosystems (1.5 vs. 0.1). In this sense it would be difficult to adopt strategies to increase resilience, since the criteria are already a part of the environment in which the agroecosystems are immersed and therefore difficult to modify.

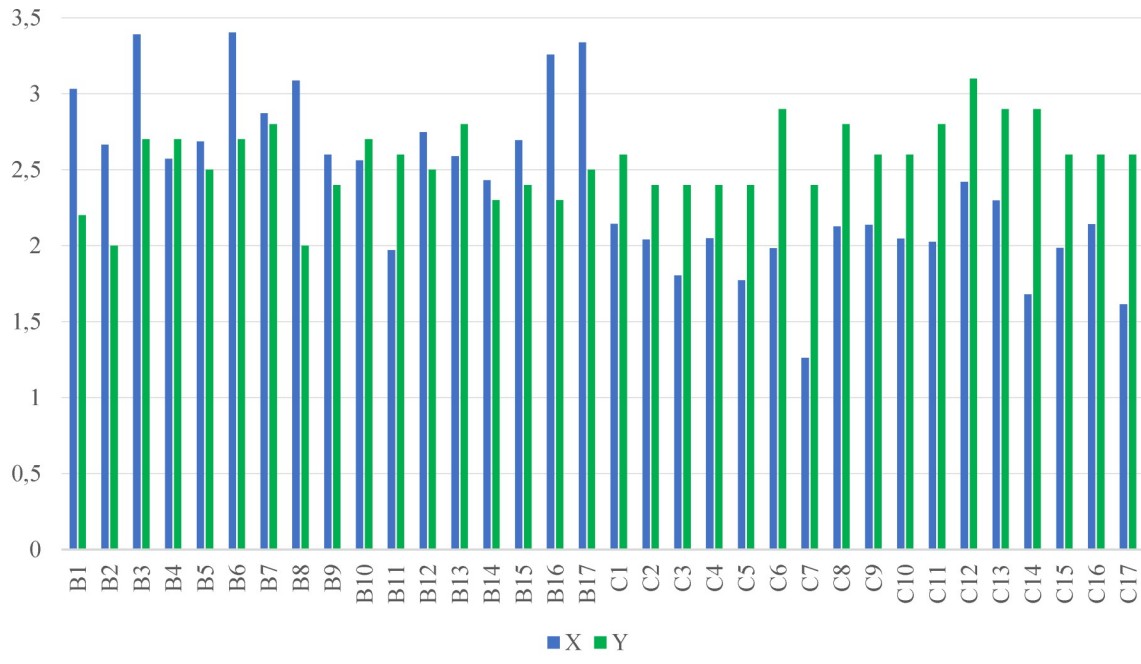

X: Weighted mean of all proposed variables Y: Mean of biophysical variables and management practices

**Fig 4. Total resilience in Brazil and Colombia comparing all proposed variables (scenario X) vs. only biophysical factors and management practices (scenario Y).**

Including all variables allows for an evaluation and analysis that can be used as an instrument to support decision-making in the short, medium and long term, as well as a tool for planning and determining effective solutions in the social sphere [104,105]. The transformations peasants require to increase their resilience involve power structures, markets, institutions and predominate societal values [101,102]. Beyond the biophysical factors and productive practices, rural populations are immersed in social contexts, within which they are challenged by political and economic differences, not only at the local scale but also at the global scale [16,18,106].

## Conclusions

The findings reveal that the level of political organization and participation in decision-making processes regarding economic, productive, technical and political components of agroecosystems, as well as the acknowledgement of rights and the determination to organize to demand them, are factors that favor the transformation of structural aspects in the municipality of Varzelandia. Therefore, the capacity of agency received a greater weight in the overall quantification of resilience.

Our attention should not only be focused on the local population's capacities to transform their conditions while understating the importance of the political, social and economic context that conditions these capacities. Conducts, values and the distribution of risks and benefits are formed by structures and social norms. Both factors are decisive in analyzing resilience.

The peasants of Marulanda have adapted to many circumstances without achieving transformation, while the peasants of Varzelandia have built effective social networks, strengthening their capacity for agency and transformation before conditions of social inequality.

The proposed methodology can be replicated in other contexts, including other indicators and weights that represent what is valued by a society, along with its knowledge and perceptions.

The proposed resilience is directed towards the formulation of strategies and policies aimed at inducing radical change at the local and regional level. In this way it cannot be constrained by access to technology or biophysical resources that favor adaptation and a limited sense of wellbeing for peasant communities.

## Supporting information

**S1 Data.**
(XLSX)

## Acknowledgments

This paper gratefully acknowledges from Interdisciplinary Center for Development Studies, Universidad de los Andes (Bogotá, Colombia); the project: "Environmental Impact Assessment in Colombia. Critical analysis and Improvement", Code Hermes: 13129, Universidad Nacional de Colombia, sede Bogotá; Der Deutsche Akademische Austauschdienst (DAAD) and The Center for Development Research (ZEF) is an institute of the University of Bonn, Germany.

## Author Contributions

**Conceptualization:** Cindy Córdoba, Catalina Triviño, Javier Toro Calderón.

**Formal analysis:** Cindy Córdoba.

**Investigation:** Cindy Córdoba, Catalina Triviño.

**Methodology:** Cindy Córdoba, Javier Toro Calderón.

**Validation:** Cindy Córdoba.

**Writing – original draft:** Cindy Córdoba, Catalina Triviño.

**Writing – review & editing:** Cindy Córdoba, Javier Toro Calderón.

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
