## [Decision Letter · Decision Letter 0]

11 Sep 2019

PONE-D-19-18560

Agroecosystem resilience. A conceptual and methodological framework for evaluation

PLOS ONE

Dear Dr Toro Calderon,

Thank you for submitting your manuscript to PLOS ONE. After careful consideration, we feel that it has merit but does not fully meet PLOS ONE’s publication criteria as it currently stands. Therefore, we invite you to submit a revised version of the manuscript that addresses the points raised during the review process.

We would appreciate receiving your revised manuscript by Oct 26 2019 11:59PM. To enhance the reproducibility of your results, we recommend that if applicable you deposit your laboratory protocols in protocols.io, where a protocol can be assigned its own identifier (DOI) such that it can be cited independently in the future. For instructions see: http://journals.plos.org/plosone/s/submission-guidelines#loc-laboratory-protocols

We look forward to receiving your revised manuscript.

Kind regards,

Juliana Hipólito, Phd

Academic Editor

PLOS ONE

Journal Requirements:

'This paper gratefully acknowledges the support and funding from Interdisciplinary Center for Development Studies, Universidad de los Andes (Bogotá, Colombia) and the project: “Environmental Impact Assessment in Colombia. Critical analysis and Improvement”, Code Hermes: 13129, Universidad Nacional de Colombia, sede Bogotá.'

'The funders had no role in study design, data collection and analysis, decision to publish, or preparation of the manuscript.'

Additional Editor Comments (if provided):

Reviewers' comments:

Reviewer's Responses to Questions

**Comments to the Author**

1. Is the manuscript technically sound, and do the data support the conclusions?

Reviewer #1: Yes

Reviewer #2: Partly

2. Has the statistical analysis been performed appropriately and rigorously? 

Reviewer #1: Yes

Reviewer #2: N/A

3. Have the authors made all data underlying the findings in their manuscript fully available?

Reviewer #1: Yes

Reviewer #2: No

4. Is the manuscript presented in an intelligible fashion and written in standard English?

Reviewer #1: Yes

Reviewer #2: Yes

5. Review Comments to the Author

Reviewer #1: This is a truly excellent contribution of the study of resiliences. The real plus comes from the methodological integration of agrarian structure and peasant community agency. My recommendation is to publish as is.

Reviewer #2: This paper purports to develop a measurement for agroecosystem resilience, making sure to integrate peasant agency and agrarian structure into a resilience measurement. The goals is to generate a measurement of resilience that centralized system transformation that are more socially just and fair, and that make smallholders and peasants more resilient. This is a laudable and interesting aim, but the paper is not completely successful, for several reasons.

(1) This paper is situated in a critique of social-ecological resilience theories, but it doesn't seem to draw upon the research that is directly engaging these questions -- transformation versus resilience, power and agency, resilience for whom and for what purposes. The paper indicates on line 114 "Resilience analyses have not considered power relations assuming the existence of a society in consensus...", but there are efforts to grapple with power relations in resilience thinking, especially coming out of political ecology or inspired by political ecological thinking (see, for instance, Cretney 2014; Leach, Scoones and Stirling 2010; Beymer Farris and Basset 2012; Walsh-Dilley 2016, Hornborg [2009] specifically addresses this question of consensus regarding resilience). I think it would make sense to situate this critique within these and other studies that are trying to do the difficult work of integrating power/agency into resilience thinking and projects.

(2) I would like a bit more work on the framework, particularly defining terms and generating a theoretical frame with which you can justify the choices you've made regarding the criteria for inclusion as well as specific variable measured (and the pathways through which you hypothesize they will impact resilience). One thing that I found curious is that the authors did not define resilience, and instead said why the usual definition from Walker does not work. What is resilience here? Also, since the authors say that there is contestation over resilience for whom?, the authors should make very clear that they are measuring peasant resilience, and particularly a vision of resilience as transformation. I also need a definition of Agrarian Structure earlier in the paper. At any rate, there must be a strong theoretical framework around resilience. Also missing was any discussion of what is expected to improve resilience or what the mechanisms are through which some variables impact resilience. The livelihoods approach might be helpful here, it seems pretty well in line with what the authors are trying to do (see McDowell and Hess 2012; Corinne Valdivia et al 2010 in the Annals). Without such a theoretical framework, I don't know where the criteria come from. Each of the variables needs to be justified, and a strong theoretical framework would be really helpful to do so. I was so curious about so many of the variables, and had no justification for why they were selected or why certain thresholds were selected.

(3) Methodologically, I'm a bit confused. The authors don't tell us that they are doing participatory research in which communities determine the relative importance of four different criteria. On one hand, this is really a lovely combination of qualitative and quantitative research -- qualitative to engage with the field sites to understand how they define resilience, and quantitative to propose a method of quanifying this insight. However, I don't quite see how this will be reproducible. Do the authors propose that this measure can be used in other cases? Or must we all go through the same qualitative process of understanding the relative importance of each criteria, as determined by members of those communities? Would this method work only for peasant communities? What might it look like in more differentiated communities? Part of the difficulty with resilience thinking is that resilience itself is contested, even how we define it is contested, but these authors don't solve that problem and seemingly focus on just peasant (although, they don't tell us if that is so, I'm reading into it.)

(4) We know very little about the two cases to which this measurement is applied. I'd like to know more about them -- size, demographics, livelihoods, products, etc. This is important to help us know if this measurement strategy could be used elsewhere.

(5) I'm confused about causality. The authors describe a transformative process for peasants in Brazil, whereby peasants gained access to land through a mobilization/resistance strategy to break of a latifundia/large landholding. This was taken as an example of resilience. But, then they are measuring resilience after this happens, and then using the protest as confirmation of high resilience.

(6) This paper would be more successful if it were framed as participatory, qualitative research showing a method for measuring resilience.

(7) What is the point of measuring resilience? In fact, resilience can't be measured ex ante because resilience is defined as an ability to adapt when bad things happen. So, by that logic, we can only measure the things that we anticipate will contribute to their capacity to do that. The authors hypothesize that greater peasant agency will allow peasants to transform unjust social situations to be more just, thereby improving their wellbeing. But the authors have done nothing to show me how that improves resilience capacity. What threat or stress does this enable them to withstand. I'm not sure this paper is about resilience at all, but more about pro-peasant transformation.

6. PLOS authors have the option to publish the peer review history of their article (what does this mean?). If published, this will include your full peer review and any attached files.

Reviewer #1: No

Reviewer #2: No

---

## [Author Response · Author response to Decision Letter 0]

11 Dec 2019

Comment 1. 

Is the manuscript technically sound, and do the data support the conclusions?

Reviewer #1: Yes

Reviewer #2: Partly

Answer 1. 

This comment does not require an answer.

Comment 2. 

¿Has the statistical analysis been performed appropriately and rigorously?

 Reviewer #1: Yes

Reviewer #2: N/A

Answer 2. 

This comment does not require an answer.

Comment 3. 

Have the authors made all data underlying the findings in their manuscript fully available?

Reviewer #1: Yes

Reviewer #2: No

Answer 3. 

The data used for the work analysis have been provided in the supporting information of the manuscript.

Comment 4. 

Is the manuscript presented in an intelligible fashion and written in standard English?

Reviewer #1: Yes; Reviewer #2: Yes

Answer 4. 

This comment does not require an answer.

Comment 5. 

Review Comments to the Author

Reviewer #1: This is a truly excellent contribution of the study of resiliences. The real plus comes from the methodological integration of agrarian structure and peasant community agency. My recommendation is to publish as is.

Reviewer #2: This paper purports to develop a measurement for agroecosystem resilience, making sure to integrate peasant agency and agrarian structure into a resilience measurement. The goals is to generate a measurement of resilience that centralized system transformation that are more socially just and fair, and that make smallholders and peasants more resilient. This is a laudable and interesting aim, but the paper is not completely successful, for several reasons.

Comment 5 (1). 

This paper is situated in a critique of social-ecological resilience theories, but it doesn't seem to draw upon the research that is directly engaging these questions -- transformation versus resilience, power and agency, resilience for whom and for what purposes. The paper indicates on line 114 "Resilience analyses have not considered power relations assuming the existence of a society in consensus...", but there are efforts to grapple with power relations in resilience thinking, especially coming out of political ecology or inspired by political ecological thinking (see, for instance, Cretney 2014; Leach, Scoones and Stirling 2010; Beymer Farris and Basset 2012; Walsh-Dilley 2016, Hornborg [2009] specifically addresses this question of consensus regarding resilience). I think it would make sense to situate this critique within these and other studies that are trying to do the difficult work of integrating power/agency into resilience thinking and projects.

Answer 5 (1). 

The comment is accepted, the adjustments to the document are: 

It is clarified that resilience analyzes are related to agroecosystems, including the respective words in red (Page 8, paragraph 1): 

“The role of organized collective agency has not been sufficiently integrated within analyses of resilience (36). More research is needed to include the ways in which human actions shape factors such as agrarian structure. Resilience analyses in agroecosystems have not considered power relations, assuming the existence of a society in consensus, …”

Recommended articles and a series of paragraphs for integrating power / agency into resilience thinking and projects were included:

Page 7, paragraphs 2-4:

“In the context of the capacity of agency, resilience has, among other aims, the achievement of food sovereignty, a concept related to the “right of communities to define their own food and agricultural systems in a culturally and ecologically appropriate way (Walsh-Dilley, Wolford W, McCarthy, 2016; Windfuhr, Jonsén, 2005). 

Resilience allows communities to have greater control over the management and governance of food and agricultural systems (Walsh-Dilley, Wolford, McCarthy, 2016), as they do not depend directly on fluctuations or disturbances of a physical, biotic, social, economic and cultural nature. Resilience is key to adaptation. It allows analyzing the need to generate changes, with planned and systematic gradual variations, giving a holding pattern to adapt to disturbances or changes, generating conditions for the dignification of the human condition.

From an agroecology approach, resilience has the following objectives: 1) to provide sufficient, healthy, nutritious and culturally appropriate food; 2) to allow consumers to decide what foods they consume and how they consume them; 3) to value and support food suppliers, with a particular focus on small family farmers and peasants; 4) to locate control and access to natural resources and guarantee democratic participation in agricultural policy decision making; 5) to protect the access of small farmers, pastoralists, fishermen and landless people to resources such as land, seeds, livestock and credit breeds; 6) to value and contribute to local knowledge and skills; 7) to value agroecology as a way to achieve environmental integrity and sustainable livelihoods (Walsh-Dilley, Wolford W, McCarthy, 2016; Windfuhr, Jonsén, 2005; Patel , 2009) and 8) to generate conditions so that systems can organize themselves (Maturana, Varela, 2003). 133 in what they call “autopoiesis” or the organization of the living (Paucar-Caceres, Harnden, 2011)”.

Page 8, paragraphs 2-4:

“However, some authors have tried to integrate factors such as power, inequality and agency into resilience thinking. Cretney (2014), McDowell and Hess (2012), Ribot (2011), Mackinnon & Derickson (2013) argue that resilience has been used to reproduce neoliberal discourses, reducing state participation, while promoting community self-sustainability and individual responsibility. In this way, hegemonic thoughts and values are normalized and multiplied. 

Contrary to the idea of resilience as a return to a state of normalcy or adaptation to unfair social conditions, another approach raises resilience as a transformation of social and environmental problems, based on forms of community activism and organization (Brown, 2011; Davidson, 2010). Articulating resilience with the fundamental problems of capitalism is still a field to explore at a theoretical and empirical level (Cretney, 2014).

Hornborg (2009), on the other hand, points out that although conflicts, inequality and power relations are part of human societies, there is a generalized assumption that there is consensus among social groups. This ignores unequal exchanges of technology, information and resources in the predominant development model. Therefore, the concept of socio-ecological resilience must address the historical problems of conflicts of interest, asymmetries of power and inequality in distribution (Friend & Moench, 2013; Pelling & Manuel-Navarrete, 2011)”. 

Page 9, paragraphs 1-3:

“However, it would not be logical for capitalism to admit or promote its antithesis, that is, a definition that questions it in its structural aspects.

Sectors with power in society seek to create a generalized approach to sustainability, which marginalizes and excludes populations with less political and economic power. In this way, the latter forge a path through citizen action and social movements to exercise governance based on principles of learning, participation and trust (Scoones and Stirling, 2010).

In the case of agroecosystems, it is necessary to consider that beyond the biophysical space they occupy, they are also influenced by symbolic, economic, political and technological relationships that interact on multiple scales and hierarchies (Cabell & Oelofse, 2012). Therefore, agroecosystems also reflect disputes over natural and economic resources and are affected by social struggles”.

A clarification is added on the page 9, paragraph 4:

“This set of factors allows farmers not only to endure any type of stress (economic, political, ecological, etc.), but to act on these factors to transform them in their favor”.

Comment 5 (2). 

I would like a bit more work on the framework, particularly defining terms and generating a theoretical frame with which you can justify the choices you've made regarding the criteria for inclusion as well as specific variable measured (and the pathways through which you hypothesize they will impact resilience). One thing that I found curious is that the authors did not define resilience, and instead said why the usual definition from Walker does not work. What is resilience here? Also, since the authors say that there is contestation over resilience for whom?, the authors should make very clear that they are measuring peasant resilience, and particularly a vision of resilience as transformation. I also need a definition of Agrarian Structure earlier in the paper. At any rate, there must be a strong theoretical framework around resilience. Also missing was any discussion of what is expected to improve resilience or what the mechanisms are through which some variables impact resilience. The livelihoods approach might be helpful here, it seems pretty well in line with what the authors are trying to do (see McDowell and Hess 2012; Corinne Valdivia et al 2010 in the Annals). Without such a theoretical framework, I don't know where the criteria come from. Each of the variables needs to be justified, and a strong theoretical framework would be really helpful to do so. I was so curious about so many of the variables, and had no justification for why they were selected or why certain thresholds were selected.

Answer 5 (2). 

The comment is accepted, the adjustments to the document are:

A proper description of the concept of resilience is included in the context of the analysis of the resilience of agroecosystems, as recommended by the reviewer:

Page 3, paragraph 3; Page 4, paragraph 1:

“To draw on the ideas of several authors (Friend & Moench, 2013; Folke et al., 2010; Davoudi, 2012; Pendall, Foster, & Cowell, 2009; Carpenter et al., 2005; Berkes & Folke, 1998; Pelling & Manuel-Navarette, 2011), resilience is proposed as an emergent property of complex systems (family, region, nation) and as a result of the dialectical multi-level interaction of its elements. This property allows the system to buffer, to adapt to and most importantly, to innovate and to be transformed, not only in response to specific points of tensions, but also to continuous and inevitable biophysical and social changes in the environment. Resilience is not a neutral concept. It is lodged within power relations and should be analyzed considering point of view, interests, access to resources, and the researchers’ place in society. Therefore, it does not only or mainly represent a system’s return to a “normal” state. Rather, on the contrary, it necessarily implies discontinuities, structural changes and dynamic developments within the system”. 

Some characteristics of resilience are confirmed for the analysis of diverse scenarios, this type of clarification or complementary concept was recommended by the reviewer:

Page 12, paragraph 2:

“Resilience cannot necessarily be measured after specific events that cause shock or stress, since when dealing with complex systems that are subject to constant change and fluctuation, it is possible to identify their degree of resilience at any time. In addition, not every change or alteration is negative, but can lead to alternative scenarios and new directions desired by the farmers. Resilience can be measured to identify crucial points needed to achieve structural transformations through public policies, management plans and demands for social struggle”.

Following the recommendations of the reviewer, the concept of agrarian structure is extended:

Page 4, paragraph 4, Page 5, paragraphs 1, 3 

“Agrarian structure is understood as a territorial configuration that includes the way in which the entire territory is distributed and occupied (Machado, 2002; Mançano-Fernandes, 2009). From this formation, power relations are established as determined by development models that promote changes mainly in the size of the property, its concentration and / or fragmentation, as well as changes in land use (Escobal & Armas, 2015). Machado (2002), describes the following configurations of agrarian land structure:

- Unimodal: - Most of the land is distributed among medium-sized landowners (low concentration of property).

- Bimodal: Most of the land is distributed among a few owners, leaving a small proportion in the hands of many small owners (high concentration of property).

- Multimodal: Small and medium-sized property dominates the structure, coexisting in turn with large, highly use property. All these are articulated through alliances, business and cooperation systems (low concentration of property)”.

“Additionally, in small portions of land it is not possible to develop traditional cultivation methods, such as rotations, patches of vegetation, fallow areas, etc”.

Page 6, paragraphs 1, 2:

“Social dynamics determine how resources are distributed among social groups and therefore those who are most exposed to risk or are most vulnerable (McDowell, 2012). Identifying the structural causes of such vulnerability makes it possible to determine strategies to increase the resilience of rural communities, enabling their access to resources. This is what the livelihoods approach tries to do by including different types of factors. These incorporate productive practices, which includes criteria such as soil management and biodiversity that allow us to identify the ways in which agroecosystems deal with issues such as climatic variability. Another group of factors involves agroecosystem conditions and context that determine their degree of vulnerability. This group includes farmers’ biophysical, social, and health criteria. Each criterion allows us to establish what resources communities have and where they stand in the face of continuously changing environmental conditions. The agrarian structure factor, although it has been mentioned by authors such as MacDowell (2012) as the most important asset for farmers, had not been quantified, nor have the criteria used for its specific analysis in agroecosystems been determined. Therefore, criteria such as productive relationships and land tenure reveal power relations that affect rural resilience and have been made invisible in agroecosystem studies. The last factor is the capacity of agency that will be developed in the next section.

 The inclusion of this set of factors gives a more complete understanding of the agroecosystem within a framework of the asymmetries of power evidenced by each of the criteria. This approach allows for the simultaneous consideration of factors at multiple levels that shape resilience as a process, not only of adaptation but of transformation, towards conditions of greater dignity”.

It is necessary to clarify that the extension of the theoretical framework on resilience was addressed in a complementary way in the response (1) that was presented in the previous section.

An analysis of resilience and agency capacity is included, as recommended by the reviewer:

Page 7, paragraphs 2-4:

“In the context of the capacity of agency, resilience has, among other aims, the achievement of food sovereignty, a concept related to the “right of communities to define their own food and agricultural systems in a culturally and ecologically appropriate way (Walsh-Dilley, Wolford W, McCarthy, 2016; Windfuhr, Jonsén, 2005)”.

Resilience allows communities to have greater control over the management and governance of food and agricultural systems (Walsh-Dilley, Wolford W, McCarthy, 2016), as they do not depend directly on fluctuations or disturbances of a physical, biotic, social, economic and cultural nature. Resilience is key to adaptation. It allows analyzing the need to generate changes, with planned and systematic gradual variations, giving a holding pattern to adapt to disturbances or changes, generating conditions for the dignification of the human condition.

From an agroecology approach, resilience has the following objectives: 1) to provide sufficient, healthy, nutritious and culturally appropriate food; 2) to allow consumers to decide what foods they consume and how they consume them; 3) to value and support food suppliers, with a particular focus on small family farmers and peasants; 4) to locate control and access to natural resources and guarantee democratic participation in agricultural policy decision making; 5) to protect the access of small farmers, pastoralists, fishermen and landless people to resources such as land, seeds, livestock and credit breeds; 6) to value and contribute to local knowledge and skills; 7) to value agroecology as a way to achieve environmental integrity and sustainable livelihoods (Walsh-Dilley, Wolford W, McCarthy, 2016; Windfuhr, Jonsén, 2005; Patel, 2009) (37–39) and 8) to generate conditions so that systems can organize themselves (40) in what they call “autopoiesis” or the organization of the living (Paucar-Caceres, Harnden, 2011)”.

Comment 5 (3). 

Methodologically, I'm a bit confused. The authors don't tell us that they are doing participatory research in which communities determine the relative importance of four different criteria. On one hand, this is really a lovely combination of qualitative and quantitative research -- qualitative to engage with the field sites to understand how they define resilience, and quantitative to propose a method of quantifying this insight. However, I don't quite see how this will be reproducible. ¿Do the authors propose that this measure can be used in other cases? Or must we all go through the same qualitative process of understanding the relative importance of each criteria, as determined by members of those communities? Would this method work only for peasant communities? What might it look like in more differentiated communities? Part of the difficulty with resilience thinking is that resilience itself is contested, even how we define it is contested, but these authors don't solve that problem and seemingly focus on just peasant (although, they don't tell us if that is so, I'm reading into it.)

Answer 5 (3) 

The comment is accepted, the adjustments to the document are:

It is clarified that the proposed methodology is participatory and the ability to adapt in other scenarios or cases:

Page 12, paragraph 5; Page 13, paragraph 1.

“This methodological approach is mixed methods and participatory, since communities and local actors are consulted with to determine both qualitative and quantitative values of factors, criteria and variables. This characteristic allows the methodology to be applied or reproduced in other territorial contexts, because the variables are evaluated according to the characteristics of the study site and are qualified according to generic scores applicable in various rural contexts (Table 1, 2). The weights are adapted to the criteria of local actors that participate, whether indigenous, black or afro communities or others”.

Comment 5 (4). 

We know very little about the two cases to which this measurement is applied. I'd like to know more about them -- size, demographics, livelihoods, products, etc. This is important to help us know if this measurement strategy could be used elsewhere.

Answer 5 (4) 

The comment is accepted, the adjustments to the document are:

Information suggested by the evaluator is included:

Page 20, paragraph 2-5:

“The municipality of Marulanda has an area of 37,857 hectares and is divided into 14 neighborhoods. Soils are predominantly susceptible to erosive processes and high slopes are prevalent. There is a wide network of rivers and streams, an average temperature of 19°C and an altitudinal range between 1550 and 3750 meters above sea level (Corpocaldas, 2001; IGAC, 2004).

In this municipality, which has an approximate population of 2,081 people (DANE, 2019) and whose Family Agricultural Unit (UAF) is defined at 18.83 hectares, the base of the economy in the agroecosystems located at an average altitude of 1800 masl is the cultivation of coffee, sugarcane cane and avocado (Unión Temporal Estudios Ambientales y Planificación Territorial río Guarinó, 2016). On the other hand, agroecosystems located at an average height of 2800 masl rely mainly on the livestock sector through dairy farming and cattle raising. There is also a small number agroecosystems dedicated to sheep raising for meat and wool (Alcaldía de Marulanda, 2016).

Varzelandia has an area of 81,499 ha, an altitude of 745 meters above sea level, an average temperature of 2 °C and an estimated population of 19320 inhabitants (IBGE, 2019). The area is predominantly dry forest which has been diminished due to the extensive livestock activities that occupy almost 77% of the municipality. These large agroecosystems dedicated to livestock belong to 8 landowners (CAA-NM, 2012).

The agroecosystems that occupy the remaining percentage of the municipality are allocated to subsistence agriculture with corn, beans and sugarcane plantations as well as small-scale livestock production. The production is for self-supply. Many of the surplus products are sold internally and others in the urban area of the municipality. The main product that is transformed is cassava flour and in some productive units, panela is produced (CAA-NM, 2012)”.

Comment 5 (5). 

I'm confused about causality. The authors describe a transformative process for peasants in Brazil, whereby peasants gained access to land through a mobilization/resistance strategy to break of a latifundia/large landholding. This was taken as an example of resilience. But, then they are measuring resilience after this happens, and then using the protest as confirmation of high resilience.

Answer 5 (5) 

The comment is accepted, the adjustments to the document are:

This point is clarified according to the recommendation:

Page 22, paragraph 1:

“It is necessary to clarify that resilience was measured at the present time, the way in which each community arrived at its current state of resilience state was analyzed and qualified, as well as the main factors that led them to this point and that determined the degree of resilience of each location”.

Comment 5 (6). 

This paper would be more successful if it were framed as participatory, qualitative research showing a method for measuring resilience.

Answer 5 (6) 

The comment is accepted, the adjustments to the document are:

It is clarified that the article proposes a complementary participatory methodology that shows a method to evaluate / measure resilience:

Page 4, paragraph 2: 

“The purpose of this article is to present a conceptual framework and complementary participatory methodology for analyzing and evaluating/measuring agroecological resilience,”

Page 12, paragraph 5:

“This methodological approach is mixed methods and participatory, since communities and local actors are consulted with to determine both qualitative and quantitative values of factors, criteria and variables”.

Comment 5 (7). 

What is the point of measuring resilience? In fact, resilience can't be measured ex ante because resilience is defined as an ability to adapt when bad things happen. So, by that logic, we can only measure the things that we anticipate will contribute to their capacity to do that. The authors hypothesize that greater peasant agency will allow peasants to transform unjust social situations to be more just, thereby improving their wellbeing. But the authors have done nothing to show me how that improves resilience capacity. What threat or stress does this enable them to withstand. I'm not sure this paper is about resilience at all, but more about pro-peasant transformation.

Answer 5 (7) 

The comment is accepted, the adjustments to the document are:

The comment is partially accepted, the article has as subject of study the agroecological resilience and the proposal of a methodology for its analysis or measurement, thus it is established from theoretical framework to the proposed and applied methodology, regarding the peasant transformation, it acts as a form of adaptation or resilience acquired.

To clarify: What is the point of measuring resilience? the following paragraph was included:

Page 12, paragraph 2:

 “Resilience cannot necessarily be measured after specific events that cause shock or stress, since when dealing with complex systems that are subject to constant change and fluctuation, it is possible to identify their degree of resilience at any time. In addition, not every change or alteration is negative, but can lead to alternative scenarios and new directions desired by the farmers. Resilience can be measured to identify crucial points needed to achieve structural transformations through public policies, management plans and demands for social struggle”.

Comment 6. 

6. PLOS authors have the option to publish the peer review history of their article (what does this mean?). If published, this will include your full peer review and any attached files.

Do you want your identity to be public for this peer review? For information about this choice, including consent withdrawal, please see our Privacy Policy.

Reviewer #1: No

Reviewer #2: No

---

## [Decision Letter · Decision Letter 1]

12 Feb 2020

PONE-D-19-18560R1

Agroecosystem resilience. A conceptual and methodological framework for evaluation

PLOS ONE

Dear Dr Toro Calderon,

Thank you for submitting your manuscript to PLOS ONE. After careful consideration, we feel that it has merit but does not fully meet PLOS ONE’s publication criteria as it currently stands. Therefore, we invite you to submit a revised version of the manuscript that addresses the points raised during the review process.

We would appreciate receiving your revised manuscript by Mar 28 2020 11:59PM. To enhance the reproducibility of your results, we recommend that if applicable you deposit your laboratory protocols in protocols.io, where a protocol can be assigned its own identifier (DOI) such that it can be cited independently in the future. For instructions see: http://journals.plos.org/plosone/s/submission-guidelines#loc-laboratory-protocols

We look forward to receiving your revised manuscript.

Kind regards,

Juliana Hipólito, Phd

Academic Editor

PLOS ONE

Additional Editor Comments (if provided):

Dear authors, despite the great progress made in this version of the article and the great contribution of this manuscript, I need to agree with the reviewer 2. There are still points related mainly to the theoretical foundation of the article that needs to be improved. I ask you to consider comments from mainly reviewer 2 for a new assessment.

Reviewers' comments:

Reviewer's Responses to Questions

**Comments to the Author**

1. If the authors have adequately addressed your comments raised in a previous round of review and you feel that this manuscript is now acceptable for publication, you may indicate that here to bypass the “Comments to the Author” section, enter your conflict of interest statement in the “Confidential to Editor” section, and submit your "Accept" recommendation.

Reviewer #1: All comments have been addressed

Reviewer #2: (No Response)

2. Is the manuscript technically sound, and do the data support the conclusions?

Reviewer #1: Yes

Reviewer #2: Yes

3. Has the statistical analysis been performed appropriately and rigorously? 

Reviewer #1: N/A

Reviewer #2: Yes

4. Have the authors made all data underlying the findings in their manuscript fully available?

Reviewer #1: Yes

Reviewer #2: Yes

5. Is the manuscript presented in an intelligible fashion and written in standard English?

Reviewer #1: Yes

Reviewer #2: No

6. Review Comments to the Author

Reviewer #1: It is fully ready to be published. This is an excellent paper, which makes important contributions to field of resilience studies by providing a reading from the perspective of agrarian structures.

Reviewer #2: Again, I applaud the inclusion of agrarian structure and peasant agency into a statistical analysis. I think that the model is well defined, interesting, and useful. However, the way that resilience is conceptualized and framed theoretical remains highly problematic. The authors write: "resilience has among other aims, the achievement of food sovereignty" (page 6), later (page 7) the authors write: "resilience allows communities to have greater control over management and governance of food & agricultural system..." and also on page 7 "resilience is key to adaptation". Then later, on page 7, the authors suggests that "from an agroecological perspective" resilience is basically the same definition of food sovereignty. This framing is problematic. I wonder if this is a language issue, but I believe the authors have confused the dependent and independent variables. Are the authors trying to measure resilience? If so, resilience is the dependent variable. They should not then be identifying what factors resilience leads to, and then measuring those factors as if they were the dependent variable, but go on to make a claim about resilience. I am more aware of research that suggests that greater peasant agency, greater control of resources, and a higher degree of food sovereignty leads to (as causal factors) higher resilience. If resilience is what we are trying to measure/explain, then we must frame resilience as the dependent, not the causal, factor. Thus, it is not correct to say that "resilience is key to adaptation", but that adaptability is the key factor that contributes to resilience. Similarly, greater control over food and agrarian systems is a factor that helps to develop (causally) resilience capacity.

The methodological and empirical parts of this paper are good. But the theoretical framework from page 2-8 remain highly problematic.

Also, as an aside, I would very much like to see an argument of the paper earlier on -- preferably in both the abstract and the introduction. I think it is something like: "Agrarian structure and peasant agency are important elements in determining resilience capacities. This paper finds that when agrarian structure is more equal, and when peasant agency is more strongly developed through political organizations, training, and involvement of women, then there is greater capacities for generating transformational resilience that improves the dignity of peasant livelihoods.

7. PLOS authors have the option to publish the peer review history of their article (what does this mean?). If published, this will include your full peer review and any attached files.

Reviewer #1: No

Reviewer #2: No

---

## [Author Response · Author response to Decision Letter 1]

25 Mar 2020

Reviewer #2: Again, I applaud the inclusion of agrarian structure and peasant agency into a statistical analysis. I think that the model is well defined, interesting, and useful. However, the way that resilience is conceptualized and framed theoretical remains highly problematic. The authors write: "resilience has among other aims, the achievement of food sovereignty" (page 6), later (page 7) the authors write: "resilience allows communities to have greater control over management and governance of food & agricultural system..." and also on page 7 "resilience is key to adaptation". Then later, on page 7, the authors suggests that "from an agroecological perspective" resilience is basically the same definition of food sovereignty. This framing is problematic. I wonder if this is a language issue, but I believe the authors have confused the dependent and independent variables. Are the authors trying to measure resilience? If so, resilience is the dependent variable. They should not then be identifying what factors resilience leads to, and then measuring those factors as if they were the dependent variable, but go on to make a claim about resilience. I am more aware of research that suggests that greater peasant agency, greater control of resources, and a higher degree of food sovereignty leads to (as causal factors) higher resilience. If resilience is what we are trying to measure/explain, then we must frame resilience as the dependent, not the causal, factor. Thus, it is not correct to say that "resilience is key to adaptation", but that adaptability is the key factor that contributes to resilience. Similarly, greater control over food and agrarian systems is a factor that helps to develop (causally) resilience capacity. 

The authors thank the reviewer for the observations offered. It is correct that the writing in one of the paragraphs on page 7 leads to an understanding of resilience as an independent variable or causal factor. Our objective in this article is to consistently identify resilience as the dependent variable. Rural communities with a greater degree of food sovereignty, greater control of resources and the capacity to manage these, as well as true participation in political processes that allow for the transformation of the conditions of inequality, can develop and/or strengthen their resilience. As a means of response, we have complemented both the introduction as well as page 7, to clarify that resilience is considered a dependent variable.

Also, as an aside, I would very much like to see an argument of the paper earlier on -- preferably in both the abstract and the introduction. I think it is something like: "Agrarian structure and peasant agency are important elements in determining resilience capacities. This paper finds that when agrarian structure is more equal, and when peasant agency is more strongly developed through political organizations, training, and involvement of women, then there is greater capacities for generating transformational resilience that improves the dignity of peasant livelihoods. 

The authors accept this comment and incorporate the ideas into the summary as well as the introduction (page 2, paragraph 1; page 4, paragraph 2; page 7, paragraph 2).

---

## [Editor Report · Decision Letter 2]

30 Mar 2020

Agroecosystem resilience. A conceptual and methodological framework for evaluation

PONE-D-19-18560R2

Dear Dr. Toro Calderon,

We are pleased to inform you that your manuscript has been judged scientifically suitable for publication and will be formally accepted for publication once it complies with all outstanding technical requirements.

With kind regards,

Juliana Hipólito, Phd

Academic Editor

PLOS ONE

Additional Editor Comments (optional):

Dear authors, I'm pleased to accept your manuscript for publication in PlosOne. I read the paper and analyzed that you consider all the reviewers' comments. I also consider that another round of reviews is no longer necessary. I only ask you to observe again submission guidelines, I found odd on page 8, the first line that you began with a number "(55) On the other hand, points out that although conflicts, inequality, and power relations are part of human societies, there is a generalized assumption that there is consensus among social groups..." I think should be something like "Hornborg (55) on the other hand..." but please check this before publication. 

Congratulations! 
---

## [Editor Report · Acceptance letter]

6 Apr 2020

PONE-D-19-18560R2 

Agroecosystem resilience. A conceptual and methodological framework for evaluation 

Dear Dr. Toro Calderón:

I am pleased to inform you that your manuscript has been deemed suitable for publication in PLOS ONE. Congratulations! Your manuscript is now with our production department. 

With kind regards,

on behalf of

Dr. Juliana Hipólito 

Academic Editor

PLOS ONE